# OpenReview forum: "Ring-A-Bell! How Reliable are Concept Removal Methods For Diffusion Models?"
_ICLR.cc/2024/Conference — ICLR 2024 poster_

### Official Review · Reviewer_nnoZ · 2023-10-30

**Soundness:** 3 good
**Presentation:** 3 good
**Contribution:** 3 good
**Rating:** 6
**Confidence:** 3

**Summary:**

This work proposed Ring-A-Bell, a model-agnostic red-teaming tool for T2I diffusion models, which serves as a prompt-based concept testing framework that generates problematic prompts to red-team T2I diffusion models with safety mechanisms.

**Strengths:**

Overall this paper proposed a practical and intersting offline method in generating problematic prompts for 'safe models'. The experiments are very convincing and concrete.

**Weaknesses:**

Please see the questions.

**Questions:**

There are a bunch of notation issues. I list some of them below:
1. What is $\rho$ in (2)? I cannot find it in the main paper.
2. What is the training parameter of (2)? Is it $\widetilde c$?
3. Are there brackets in (3)
4. Should $\tilde{\mathbf{P}}_{cont}$ be a function of $c$ or $\hat c$?

Question about experiments:
1. How to tell whether the percentage of nudity is greater than 50%? The output propobility? Please be rigorous.

---

> ### Author Response · Authors · 2023-11-20
> **Response for Reviewer nnoZ**
>
> We appreciate the review for providing feedback such as “*The experiments are very convincing and concrete...*”. We are content that the reviewer shares common concerns on the safety of T2I diffusion models. We further address the reviewer’s comment in the following.
>
> &nbsp;
>
> ### Notation Issues
> ----
>
>
> Thanks for pointing out the crucial point, we will improve the notation in the revised version, marked in blue. Furthermore, we will include a notation table in Appendix G for the reviewer’s reference. Below we will list the responses regarding notations in a pointwise manner.
> * $\rho$ denotes the weight of the loss between the original and the modified model.
> * Indeed, the training objective is to search for an adversarial concept $\tilde{c}$ such that the loss is minimized, i.e., $\tilde{c} := \arg\min_{\tilde{c}} L_{white}$.
> * There should be brackets in (3) as we are extracting the concept information by contrasting two semantically similar prompts.
> * Since we have no true latent vector of $c$, when producing $\mathbf{\tilde{P}}_{cont}$ we instead opt for the empirical representation $\hat{c}$ for attack. Therefore, it is a function of $\hat{c}$.
>
> &nbsp;
>
> ### Nudity Percentage of I2P Prompts
> ----
>
> According to the dataset description [R1], the percentage of nudity is an attribute in the I2P dataset, which is the percentage of images depicting explicit nudity as per the NudeNet classifier/detector out of 10 generated images using Stable Diffusion.
>
> &nbsp;
>
> [R1]: Artificial Intelligence & Machine Learning Lab at TU Darmstadt. “AIML-Tuda/I2P · Datasets at Hugging Face.” AIML-TUDA/I2p · Datasets at Hugging Face, 20 Nov. 2022, huggingface.co/datasets/AIML-TUDA/i2p.

---

> ### Author Response · Authors · 2023-11-22
> **Looking forward to reviewer nnoZ’s feedback**
>
> &nbsp;
>
> We appreciate the positive feedback and constructive comments from reviewer nnoZ in the initial reviews. As the discussion deadline nears, we haven't received additional feedback on our responses. We intend to use OpenReview's interactive feature to engage in discussions with the reviewer, confident that our response adequately addresses your concerns. Specifically, we've organized our response according to the reviewer’s suggestion as the following:
>
> * Notation Issues
> * Explanation of Nudity Percentage of I2P Prompts
>
> We hope our responses convince the reviewer about the merits of this work. If the reviewer has any other suggestions or comments, please don't hesitate to let us know!
>
> Best Regards, \
> Authors of Ring-A-Bell

---

> > ### Comment · Reviewer_nnoZ · 2023-12-04
> >
> > I have checked the responses as well as comments from other reviewers. I will keep my score towards acceptance.

---

### Official Review · Reviewer_nmRw · 2023-10-31

**Soundness:** 2 fair
**Presentation:** 3 good
**Contribution:** 2 fair
**Rating:** 8
**Confidence:** 5

**Summary:**

This paper investigates the safety of text-to-image models. The paper proposes a model-agnostic attack to evade safety mechanisms and generate sensitive and inappropriate images. The proposed work is evaluated on online services to explore their safety risks.

**Strengths:**

1. The paper investigates red-teaming text-to-image models, which is a critical topic for generative AI safety.
2. The proposed method is validated on four T2I online services.
3. The paper is well-written and easy to follow.

**Weaknesses:**

1. The model-agnostic design of the proposed framework is not convincing. The entire design is based on an offline CLIP model and is irrelevant to the online services. This design implicitly assumes that the framework that applies to the offline CLIP model can be transferable and effective for online services. What if the online services use a more robust text encoder? In addition, one of the contributions, claimed by the paper, is that Ring-A-Bell is “based solely on either the CLIP model or general text encoders.” However, in the evaluation, only the CLIP model is evaluated. It would be great to see if the proposed work can be extended to other and more recent text encoders.
2. The paper only compares the proposed framework with QF-Attack, which is insufficient. Many recent works are encouraged to be investigated [1-4]. In addition, although P4D is designed for offline attacks, it would be great to consider P4D as a baseline to compare the performance gap between online and offline attacks.
3. The paper aims to evade the safety mechanism of online diffusion models. However, the paper only considers concept removal defenses. For an online service, an easy and effective way is to develop a detector to identify inappropriate images. For example, the service provider could build a detector (e.g., NudeNet detector used in the evaluation) to detect nudity in the images. The proposed framework that mainly focuses on the text domain may not be effective.

[1] Qu, Yiting, Xinyue Shen, Xinlei He, Michael Backes, Savvas Zannettou, and Yang Zhang. "Unsafe diffusion: On the generation of unsafe images and hateful memes from text-to-image models." ACM CCS 2023.
[2] Mehrabi, Ninareh, Palash Goyal, Christophe Dupuy, Qian Hu, Shalini Ghosh, Richard Zemel, Kai-Wei Chang, Aram Galstyan, and Rahul Gupta. "Flirt: Feedback loop in-context red teaming." arXiv preprint arXiv:2308.04265 (2023).
[3] Rando, Javier, Daniel Paleka, David Lindner, Lennart Heim, and Florian Tramèr. "Red-teaming the stable diffusion safety filter." arXiv preprint arXiv:2210.04610 (2022).
[4] Yang, Yuchen, Bo Hui, Haolin Yuan, Neil Gong, and Yinzhi Cao. "SneakyPrompt: Evaluating Robustness of Text-to-image Generative Models' Safety Filters." arXiv preprint arXiv:2305.12082 (2023).

**Questions:**

Please clarify the model-agnostic design and explain why it is effective for online services.


The rebuttal has addressed most of my concerns.

---

> ### Author Response · Authors · 2023-11-20
> **Response for Reviewer nmRw -- Part I**
>
> Thanks for your genuine appreciation of the clarity and precious reviews of our work! We are delighted to receive a comment denoting that “*The paper investigates red-teaming text-to-image models, which is a critical topic for generative AI safety.*” Please see the response below as we address your comments.
>
> &nbsp;
>
>
> ### Transferability and Model-Agnostic Design
> ---
> We thank the reviewer for raising the question. Indeed we’ve attempted to expand Ring-A-Bell to other possible text encoders such as BERT and T5. However, there are certain critical differences that hinder such encoders from achieving comparable performance against the CLIP-Based Ring-A-Bell. The main reason lies in that these text encoders specialize **in the text domain and construct respective embedding space, while the embedding space for CLIP is much more aligned from the vision-language aspect.** Secondly, the hard prompt provided by Ring-A-Bell starts from an original soft prompt specific to CLIP with further discrete optimization. This model-specific conversion could hinder the generated prompt to transfer under various text encoder settings.
>
> Nevertheless, we note that while the transferability is limited, in practice, **there is no reason to restrain the attacker to only use one text encoder such as CLIP.** We do not consider the differences in transferability as a weakness. **As long as the attacker can find one prompt to sabotage the target model, the attack is considered successful.** Therefore, the more text encoders the attacker can use, the more diverse prompts the attacker can create to test the target model.
>
> Moreover,  we note that most online T2I services would reveal not all but certain information about the model architecture or the underlying concept. This allows us to obtain the corresponding text encoder information for Ring-A-Bell to perform concept extraction. On the other hand, Ring-A-Bell could as well actively extract copies of certain concepts with possible text encoder candidates and apply the discrete optimization to acquire the respective hard prompt for evaluation.
>
> **Under this aspect, as long as a single problematic prompt from any text encoder has succeeded, this signifies the security breach of the corresponding of T2I models and its subsequent risk. Therefore we argue that the transferability of Ring-A-Bell does not hinder its potential in seeking possible risk of T2I models.**
>
> As for the model-agnostic design, we note that since most of the concept removal methods fine-tune the diffusion model itself and simply fix the text encoder, the concept might not be certifiably removed from the diffusion model. In fact, one of the recent studies demonstrated through causal mediation analysis that the **text encoder has a certain extent of participation in generating the images** [R8]. Therefore, we postulate that by **forming a holistic empirical representation $\hat{c}$** we are able to leverage the remnant part of the concept, using the knowledge of text encoders, to produce inappropriate images.
>
> &nbsp;
>
> ### Concept Removal and Beyond
> ---
> As for the comment of “only evaluating the concept removal methods”, we believe that the reviewer has confused some of our experiment settings in the evaluation. We here clarify that since Ring-A-Bell requires only black-box access to the T2I diffusion model, the attack applies to both concept removal methods and online services with safety filters. Particularly, we note that current online services have a content policy that includes safety measures [R1, R2, R3], some possessing both input and output detection. Therefore, evaluating the online T2I services is equivalent to circumventing the safety filtering implemented.
>
> To further elaborate, our experiment evaluated three settings. The first one is naturally the online T2I service that **employs safety filtering**, evaluated in Section 4.1. Secondly, we evaluated existing concept removal methods **without safety filters** to demonstrate the risk and unreliability of such methods. Lastly, as a simple defense and simulation of possible scenarios, we evaluated the setting where concept removal is used in tandem **with safety filters** to show the efficacy of Ring-A-Bell in Section 4.2.
>
> &nbsp;
>
> [R1] Ultimate guide to DALL·E 2: how to use it & how to get access. (2022, July 1). DALL·Ery GALL·Ery. Retrieved November 19, 2023, from https://dallery.gallery/dall-e-ai-guide-faq/ \
> [R2] Midjourney community guidelines. (n.d.). Midjourney Documentation. Retrieved November 19, 2023, from https://docs.midjourney.com/docs/community-guidelines \
> [R3] Frequently Asked Questions. (n.d.). Runway Help Center. Retrieved November 19, 2023, from https://help.runwayml.com/hc/en-us/categories/21663959852435-Frequently-Asked-Questions \
> [R8] Basu, Samyadeep, et al. "Localizing and Editing Knowledge in Text-to-Image Generative Models." arXiv preprint arXiv:2310.13730 (2023).

---

> ### Author Response · Authors · 2023-11-20
> **Response for Reviewer nmRw -- Part II**
>
> &nbsp;
>
> ### Investigation with Related Works -- Part I
> ---
> We appreciate the reviewer for bringing up related work for discussion and investigation. Furthermore, we note that some references [R4, R6] have been discussed in Appendix B. Nevertheless, in the following, we would discuss the related work in order. Meanwhile, we will include the discussion in Appendix M for reviewer's reference.
>
> Firstly, [R4] evaluates the safety of T2I models in an exploratory manner. By manually collecting unsafe prompts from online forums, the authors examine the risk of T2I models generating inappropriate images using these collected prompts. On the other hand, the authors also trained a customized safety filter that superseded most filters deployed in online T2I services. Meanwhile, they take a step forward by aiming to fine-tune T2I models such that the model could generate hateful memes, a specific type of unsafe content in images. **While we appreciate the exploratory analysis of [R4], we note that this differs from our direction as the manually collected unsafe prompts cannot scale to provide an overall examination of the T2I model. On the other hand, we’ve already considered a similar methodology such as the I2P dataset. These manually collected prompts generally would not pass the safety filter (such as the *"Original Prompt"* row for I2P in Table 2, Section 4) but serve as a great starting template for Ring-A-Bell**.
>
> **Secondly, we note that [R5] is a very recent and even concurrent submission published in August 2023 with a different problem setup.** Specifically, [R5] aims to develop a red-teaming tool of T2I diffusion models by leveraging the power of language models (LM). Specifically, the attack is set up as a feedback loop between the language model and the T2I model. That is to say, the LM would first initiate an adversarial prompt as an input to the T2I model. Meanwhile, the output image would go through a safeness classifier and the score would serve as feedback for the LM to adjust the adversarial prompt for subsequent trials. **While the method in [R5] serves as an important red-teaming tool for T2I models, we note that current online services would directly reject the generated inappropriate image, implying no meaningful feedback could be obtained by the LM. As a result, the extension to online T2I services remains unclear and therefore differs from the setting of Ring-A-Bell.**
>
> Thirdly, [R6] aims to explore the potential risk of safety filters deployed by Stable Diffusion. Particularly, the authors proposed prompt dilution to dilute sensitive prompts such that it could circumvent the safety filtering of Stable DIffusion. **Here we note that we indeed incorporated the method of prompt dilution when evaluating Ring-A-Bell for online T2I services to increase the overall attack success rate. However, we’ve included an ablation study between the effect of Ring-A-Bell with and without dilution in Appendix H to demonstrate that simply using prompt dilution alone is not effective in constructing a successful attack.**
>
> &nbsp;
>
>
> [R4] Qu, Yiting, Xinyue Shen, Xinlei He, Michael Backes, Savvas Zannettou, and Yang Zhang. "Unsafe diffusion: On the generation of unsafe images and hateful memes from text-to-image models." ACM CCS 2023.  \
> [R5] Mehrabi, Ninareh, Palash Goyal, Christophe Dupuy, Qian Hu, Shalini Ghosh, Richard Zemel, Kai-Wei Chang, Aram Galstyan, and Rahul Gupta. "Flirt: Feedback loop in-context red teaming." arXiv preprint arXiv:2308.04265 (2023).  \
> [R6] Rando, Javier, Daniel Paleka, David Lindner, Lennart Heim, and Florian Tramèr. "Red-teaming the stable diffusion safety filter." arXiv preprint arXiv:2210.04610 (2022).

---

> ### Author Response · Authors · 2023-11-20
> **Response for Reviewer nmRw -- Part III**
>
> &nbsp;
>
> ### Investigation with Related Works -- Part II
> ---
>
> Lastly, [R7] attempts to attack the safety filter of existing online T2I services via reinforcement learning. Specifically, SneakyPrompt [R7] would initialize a target prompt and replace the sensitive tokens within. Meanwhile, the bypass-or-not response from the T2I services then serves as feedback to the agent to replace more suitable tokens until the safety filter is bypassed and the CLIP score between the target prompt and generated image is optimized. In the section below, we’ve included the comparison between SneakyPrompt and Ring-A-Bell on nudity.
>
> We conduct the official code of SneakyPrompt and follow their default setting that uses reinforcement learning to search, the CLIP score as a reward, the early stopping threshold score for the agent is 0.26, and the upper limit for query is 60. The result is demonstrated in the table below.
>
>
> &nbsp;
>
> | Attack Success Rate (ASR) | ESD        |  SLD-Max   | SLD-Strong | SLD-Medium |
> |:-------------------------:| ---------- |:----------:|:----------:|:----------:|
> |       SneakyPrompt        | 12.63%     |   3.16%    |   7.37%    |   27.37%   |
> |        Ring-A-Bell        | **35.79%** | **42.11%** | **61.05%** | **91.58%** |
>
>
> As shown in the table, the performance of SneakyPrompt is **much lower than** Ring-A-Bell in terms of concept removal methods.  This is mainly due to the fact that SneakyPrompt focuses on the jailbreak of safety filters, rendering it **unable to find the problematic prompts for concept removal methods** so as to generate inappropriate images. Specifically, if the feedback from safety filters indicates that the image is safe, SneakyPrompt will deem this prompt as successfully jailbroken. However, since concept removal methods have already **eliminated a large portion of the sensitive concept**, even if the prompt contains sensitive words, under the setting of concept removal, **the generated image is deemed safe by the safety filter.** **As a result, SneakyPrompt skips it.**
>
> &nbsp;
>
> [R7] Yang, Yuchen, Bo Hui, Haolin Yuan, Neil Gong, and Yinzhi Cao. "SneakyPrompt: Evaluating Robustness of Text-to-image Generative Models' Safety Filters." arXiv preprint arXiv:2305.12082 (2023).

---

> ### Author Response · Authors · 2023-11-20
> **Response for Reviewer nmRw -- Part IV**
>
> &nbsp;
>
> ### Comparison Against P4D
> ---
> **Disclaimer: The link might contain sensitive & inappropriate images, reader discretion is advised.**
>
> Before demonstrating the online/offline gap between P4D and Ring-A-Bell, we would like to clarify the difference in evaluation for both works. The attack success rate metric (ASR) of P4D employs **the NudeNet classifier** [R9] (binary classifier for safe/unsafe) with an unsafe threshold of 0.45 to judge whether an image is inappropriate. On the other hand, our metric employs **the NudeNet detector** [R9] that detects specific exposure parts, and flags certain parts such as the anus, and genitalia as nudity concepts.
>
> To establish the comparison against both evaluation metrics, we provide an example image that is considered generally safe with the following link (https://imgur.com/KEC7xLH). It can be seen that our results are more restricting since we recognize only certain parts as nudity-related while images such as the one in the link are recognized as unsafe in their much more lenient aspect.
>
> To provide the comparison, we conduct the official code of P4D-$N$ using prompt length as $16$ and apply an identical random seed as the one used in Ring-A-Bell. We perform evaluation under both metrics and present the table below. As one can observe from the table, the performance of P4D-$N$ **degrades heavily** under our metric and **does not stand out significantly** in comparison to our approach. On the other hand, using P4D’s metric, we also perform **superior to P4D** under the majority of different settings (3 out of 5), demonstrating the effectiveness of our method. We’ve included this result in Appendix K for the reviewer’s reference.
>
> &nbsp;
>
> | P4D-$N$ / Ring-A-Bell |         ESD         |       SLD-Max       |     SLD-Strong      |      SLD-Medium      |        SD-NP        |
> |:-----------------:|:-------------------:|:-------------------:|:-------------------:|:--------------------:|:-------------------:|
> |  Our ASR Metric   | 21.05% / **35.79%** | 12.63% / **42.11%** | 10.53% / **61.05%** | 28.42% / **91.58%** |  2.11% / **34.74%**  |
> |  P4D ASR Metric   | **66.31%** / 55.79% | **71.58%** / 57.89% | 77.89% / **86.32%** |  85.26% / **100%**  | 22.11% / **49.47%** |
>
> &nbsp;
>
> [R9] Bedapudi, P. (2021, December 9). NudeNet: An ensemble of Neural Nets for Nudity Detection and Censoring. Medium. Retrieved September 1, 2023, from https://praneethbedapudi.medium.com/nudenet-an-ensemble-of-neural-nets-for-nudity-detection-and-censoring-d9f3da721e3

---

> ### Author Response · Authors · 2023-11-22
> **Looking forward to reviewer nmRw’s feedback**
>
> &nbsp;
>
> We appreciate the positive feedback and constructive comments from reviewer nmRw in the initial reviews. As the discussion deadline nears, we haven't received additional feedback on our responses. We intend to use OpenReview's interactive feature to engage in discussions with the reviewer, confident that our response adequately addresses your concerns. Specifically, we've organized our response according to the reviewer’s suggestion as the following:
>
> * Transferability and Model-Agnostic Design
> * Concept Removal and Beyond
> * Investigation with Related Works (including experimental results)
> * Experiments on comparison against P4D (including experimental results)
>
> We hope our responses convince the reviewer about the merits of this work. If the reviewer has any other suggestions or comments, please don't hesitate to let us know!
>
> Best Regards, \
> Authors of Ring-A-Bell

---

### Official Review · Reviewer_WiiG · 2023-11-02

**Soundness:** 3 good
**Presentation:** 3 good
**Contribution:** 3 good
**Rating:** 5
**Confidence:** 5

**Summary:**

The paper investigates the effectiveness of safety mechanisms for text-to-image (T2I) diffusion models. It proposes a model-agnostic evaluation tool called Ring-A-Bell, which can assess the reliability of deployed safety mechanisms without prior knowledge of the target model. The tool performs concept extraction to identify problematic prompts and generates inappropriate content to evaluate the safety measures. The paper empirically validates the method by testing online services and various concept removal methods.

**Strengths:**

+ significance: this paper reveals the importance of adversarial evaluation of the current concept removal works. Moreover, it performs its attack in a practical black-box way, which expands its evaluation scale to commercial APIs. How to evaluate the black-box commercial APIs is of great importance since they are powerful and more easily accessible by common people.

+ quality: this paper takes a comprehensive inspect into the safety robustness of commercial APIs and also state-of-the-art concept removal methods, which demonstrates the efficacy of their method.

**Weaknesses:**

- their attack is easy to be filtered or removed by advanced NLP techniques such as large language model, since they perform token-level optimization on the prompt and the output is usually random combinations of tokens. large language model can purify the prompt by removing the semantically unclear part of the prompt.

- the token level optimization is uninterpretable and cannot provide insights into how to defend against such attacks.

**Questions:**

- the ablation study of interference among modification, prompt dilution, and Ring-A-Bell: in Figure 2 and 3, the shown prompt contains the three types of texts. 1) is the whole prompt generated by Ring-A-Bell, or you combine the three types of attacks together for final output? how to discriminate the type of texts such as modification, prompt dilution, and Ring-A-Bell? 2) which type of text is essential to evade the safety filter of diffusion models or defeat the concept removal methods?

- the ESD config is missing in Table 2 since it has multiple variants, whose concept removal effect is different from each other.

- why K=16 is the final config? is there experiment result about smaller K?

---

> ### Author Response · Authors · 2023-11-20
> **Response for Reviewer WiiG -- Part I**
>
> We sincerely thank the reviewer for providing detailed comments on Ring-A-Bell. We appreciated the comment saying that “*this paper reveals the importance of adversarial evaluation …. Moreover, it performs its attack in a practical black-box way, which expands its evaluation scale to commercial APIs.*” and share a common perspective as the reviewer for the importance in evaluating these commercial APIs. Below we provide a pointwise reply to the reviewer’s comment.
>
> &nbsp;
>
> ### Ablation Study of Three Attack Strategies
> ---
> **Disclaimer: The link might contain sensitive & inappropriate images, reader discretion is advised.**
>
> We present the visualized ablation study in a figure with the following link (https://imgur.com/L7ObpnT). Specifically, in the figure, "Ring-A-Bell" represents our execution of the Ring-A-Bell method based on the target prompt to generate a problematic prompt. We note that the example is produced by DALL·E 2. Furthermore, in the figure, the top row represents the images by applying only modification and dilution while the bottom row applies all Ring-A-Bell, modification, and dilution techniques.
> As one can see in the figure, using approaches such as modification and dilution could allow us to **increase the overall success rate rather than only using Ring-A-Bell**. To explain the two strategies, modification simply avoids inappropriate words in the problematic prompt (input filtering), while dilution prevents generated images from being identified as inappropriate (output filtering). It's worth noting that when only using the original prompt along with the above two techniques, e.g., modification and dilution, the generated images fail to generate nudity content.
>
> That is to say, **simply using the original prompt and these two techniques does not produce inappropriate images. Problematic images would appear only when combining these techniques along with prompts generated by Ring-A-Bell.** For the reviewer’s reference, we will include the discussion in Appendix I.
>
> &nbsp;
>
> ### ESD Configuration
> ---
> For the ESD configuration, we apply the officially released pre-trained model (ESD-Nudity), which only tunes non-cross-attention parameters and sets the negative guidance as $1$. For the ESD-Violent model, we trained it based on the official code and simply fine-tuned cross-attention parameters with negative guidance set as $3$. We have included these implementation details in Section 4.
>
>
> &nbsp;
>
> ### Experiment on Smaller $K$
> ---
> We follow the settings of Table 3 in Section 4.3 to produce results for smaller $K$. Particularly, we set $K=8$ and $\eta=3$ for the nudity concept. As shown in the table below, it is clear that the results obtained with $K=8$  **are not superior to** those achieved with $K=16$. Thus, it can be inferred that small values of $K$ are not conducive to effectively causing the model to generate inappropriate images. We’ve included the additional result in Appendix J for the reviewer’s reference.
>
> &nbsp;
>
> | Attack Success Rate (ASR) | SD         | ESD        |  SLD-Max   | SLD-Strong | SLD-Medium |   SD-NP    |     CA     |    FMN     |
> |:-------------------------:| ---------- | ---------- |:----------:| ---------- |:----------:|:----------:|:----------:|:----------:|
> |           $K=8$           | 87.37%     | 14.74%     |   21.05%   | 44.21%     |   87.37%   | **37.89%** |   85.26%   |   66.32%   |
> |          $K=16$           | **93.68%** | **35.79%** | **42.11%** | **61.05%** | **91.58%** |   34.74%   | **89.47%** | **68.42%** |

---

> ### Author Response · Authors · 2023-11-20
> **Response for Reviewer WiiG -- Part II**
>
> &nbsp;
>
> ### Defense against Ring-A-Bell
> ---
> (*A large language model can purify the prompt by removing the semantically unclear part of the prompt.*)
>
> We thank the reviewer for bringing up the discussion of devising mitigation strategies against our proposed method. However, we would like to point out the inevitable competing nature of attacks and defenses and their adaptive versions, as discussed in “On Adaptive Attacks to Adversarial Example Defenses (NeurIPS)” [R6]. We would like to further articulate the following points to explain why devising reliable defenses may not be trivial:
>
> * While it is possible that some additional text filtering schemes (e.g., a perplexity-based detector or a large language model) can mitigate the effectiveness of our approach, in the “adaptive attack setting” once the bad actor has some knowledge of the deployed defenses, advanced attacks can be carried out to bypass these defenses. For example, large language models are known to suffer from prompt injection attacks [R1, R2]. The bad actor can utilize prompt injection to counter these deployed defenses. Moreover, the perplexity-based defense was shown to be effective against early-day jailbreak prompts [R3], but it was quickly shown to be broken by semantic jailbreak prompts such as [R4]. Therefore, **advanced NLP techniques may not be a long-lasting solution, unless they can be proven certifiably reliable**.
>
> &nbsp;
>
> * It is reasonable to assume many of the tested online T2I services such as Midjourney, DALL·E 2, Gen-2, and Stable Diffusion XL, should have embedded a certain level of defenses to mitigate the possibility of generating inappropriate content. For example, it is mentioned in DALL·E 2 that both text prompt input and the generated image would go through safety filtering [R5]. However, regardless of the actual mitigation strategy, **our model-agnostic method can still find many successful prompts that bypass the implemented defenses (if any) embedded in these black-box models.** The model-agnostic nature of our method enables easy testing of any defense by considering the defense as part of the target model.
>
> &nbsp;
>
> * Thirdly, we’ve included the experiment for defense such as GPT purification in the table below. Specifically, we adopt the target concept as violence and instruct ChatGPT using the template of “*Please remove the semantically unclear part of the following prompt: <original prompt>*” with *<original prompt>* being the problematic prompt produced by Ring-A-Bell. As shown in the table, it is evident that such defense **does not incur a huge drop in attack success rate (ASR)**, which signifies the effectiveness of Ring-A-Bell. Furthermore, we can also spot that under certain settings such as ESD and SLD-Max, the defense even raised the ASR of the attack, which establishes that simple defense itself does not suffice for mitigating Ring-A-Bell.
>
> &nbsp;
>
> * Lastly, even by mitigating the above issues and further establishing a certifiable defense strategy, this alone does not stand for **the certification of concept removal methods** as **simply blocking the input to be semantically correct does not imply that a certain concept has been fully removed from the diffusion model**, leaving an inherent issue.
>
> &nbsp;
>
> | Attack Success Rate (ASR) | SD        | ESD       |  SLD-Max  | SLD-Strong | SLD-Medium |  SD-NP  |    CA     |    FMN    |
> |:-------------------------:| --------- | --------- |:---------:| ---------- |:----------:|:-------:|:---------:|:---------:|
> |       GPT Purified        | 94%       | **59.2%** | **23.2%** | 45.6%      |   70.4%    |  73.2%  |   96.8%   | **80.4%** |
> |         Original          | **96.4%** | 54%       |   19.2%   | **50%**    | **76.4%**  | **80%** | **97.6%** |   79.6%   |
>
> &nbsp;
>
> [R1] Greshake, Kai, et al. "Not what you’ve signed up for: Compromising Real-World LLM-Integrated Applications with Indirect Prompt Injection." arXiv preprint arXiv:2302.12173 (2023).\
> [R2] Zou, Andy, et al. "Universal and transferable adversarial attacks on aligned language models." arXiv preprint arXiv:2307.15043 (2023).\
> [R3] Jain, Neel, et al. "Baseline defenses for adversarial attacks against aligned language models." arXiv preprint arXiv:2309.00614 (2023).\
> [R4] Liu, Xiaogeng, et al. "Autodan: Generating stealthy jailbreak prompts on aligned large language models." arXiv preprint arXiv:2310.04451 (2023).\
> [R5] https://dallery.gallery/dall-e-ai-guide-faq/ \
> [R6] Tramer, Florian, et al. "On adaptive attacks to adversarial example defenses." Advances in neural information processing systems 33 (2020): 1633-1645.

---

> ### Author Response · Authors · 2023-11-22
> **Looking forward to reviewer WiiG’s feedback**
>
> &nbsp;
>
> We appreciate the positive feedback and constructive comments from reviewer WiiG in the initial reviews. As the discussion deadline nears, we haven't received additional feedback on our responses. We intend to use OpenReview's interactive feature to engage in discussions with the reviewer, confident that our response adequately addresses your concerns. Specifically, we've organized our response according to the reviewer’s suggestion as the following:
>
> * Discussion of Defense against Ring-A-Bell (including experimental results)
> * Ablation Study of Three Attack Strategies  (including experimental results)
> * Description of ESD Configuration in Experiments
> * Experiment on Smaller $K$ values
>
> We hope our responses convince the reviewer about the merits of this work. If the reviewer has any other suggestions or comments, please don't hesitate to let us know!
>
> Best Regards,\
> Authors of Ring-A-Bell

---

### Official Review · Reviewer_UevV · 2023-11-03

**Soundness:** 2 fair
**Presentation:** 3 good
**Contribution:** 3 good
**Rating:** 5
**Confidence:** 3

**Summary:**

This paper proposed a model-agnostic red-teaming tool, Ring-A-Bell, for the evaluation of text-to-image (T2I) diffusion models’ safety mechanisms. In the first stage, this concept retrieval algorithm would perform concept extraction by learning the difference between the embeddings of prompts with/ without the target concept (e.g., violence). With the extracted concept, the algorithm utilises genetic algorithms to produce problematic prompts to test the reliability of online T2I diffusion models.

**Strengths:**

1.	The main idea of the algorithms is clearly demonstrated with figures and examples. The motivation of the paper is clearly explained by analyzing the drawbacks of the current model-specific attack algorithms.

2.	Extensive experiments are well-designed to show the efficiency of Ring-A-Bell in generating problematic prompts in the field of nudity and violence. The evaluation is reasonable with the NudeNet detector. The results are clearly shown with quantitative tables and well-processed images to demonstrate the ability of Ring-A-Bell as a red-teaming tool.

**Weaknesses:**

1. The related work should include the introduction of the concept removal methods, such as the Safe Latent Diffusion mentioned in the paper.

2. In the concept extraction stage, the selection/ generation of the prompt pairs, which are semantically similar but different from the target concept, is not clearly specified. Producing high-quality prompt pairs requires extensive specialized knowledge. This can affect the effectiveness of the algorithm and increase the difficulty of reproduction.

3. The generation of p ̃_cont is simply by a linear combination of the embedding of P and extracted empirical representation c ̂, which needs further justification. The definition of ‘target prompt P’ is not specified.

4. The ablation study is not properly implemented. For example, it might be better to demonstrate the performance of the algorithm with and without discrete optimization.

5. The algorithm strongly emphasizes that the text encoder is the CLIP model. It might be better to test on other text encoders.

**Questions:**

see the above.

---

> ### Author Response · Authors · 2023-11-20
> **Response for Reviewer UevV -- Part I**
>
> We genuinely appreciate the reviewer for the comprehensive comments concerning online T2I diffusion models. We are delighted to receive the positive feedback that “*Extensive experiments are well-designed to show the efficiency of Ring-A-Bell in generating problematic prompts in the field of nudity and violence…*” Please see our point-to-point response to your comments below.
>
> &nbsp;
>
> ### Introduction to Concept Removal Methods
> ---
> We apologize for not making the references to concept removal methods clear in the related work section due to page limits. Nevertheless, we’ve actually included a detailed related work in Appendix B, including the introduction to concept removal methods and related safety filters. In our revision, **we’ve added a line of reference in Section 2, marked in blue, in accordance with the suggestion of the reviewer**.
>
> &nbsp;
>
> ### Explanation of Prompt-Pair Selection and Generation
> ---
> Thanks for the great question. We would like to clarify that producing high-quality prompt pairs does not require extensive specialized knowledge from humans, and we simply apply existing large language models (e.g., ChatGPT) to generate such prompts.  **Specifically, for the generation of prompt-pairs, we utilize ChatGPT to create sentences about a particular concept $c$, i.e., $P_{i}^{c}$. Furthermore, when seeking semantically similar prompts without the concept, i.e., $P_{i}^{\not{c}}$, we instruct ChatGPT to retain most words in the sentences and only modify a few words related to the specific concept, preventing the need of extensive knowledge**.
>
> For instance, regarding objects or artistic styles like Van Gogh style, we ask ChatGPT to generate several words related to landscapes or natural scenery and append "with Van Gogh style" after each prompt. On the other hand, for $\not{c}$, excluding "with Van Gogh style" suffices.
>
> As for general and aggregated concepts such as nudity, we instruct ChatGPT to generate some vocabularies about nudity, such as exposed, bare, and topless. Furthermore, we define subjects and scenarios such as man, woman / bedroom, in a painting. Lastly, we ask ChatGPT to permute and construct sentences using these words. On the other hand, for $\not{c}$, simply replacing the previous sensitive words would suffice.
>
> We have added the above discussion in Appendix H to make our prompt generation process more clear.
>
> &nbsp;
>
> ### Target Prompt and the Linear Combination
> ---
> Thanks for the question regarding linear combinations. We clarify that here the target prompt implies the initial prompt that is problematic but is unable to pass the safety filters or the concept removal methods, thus rendering these prompts incompetent in generating inappropriate images. In the experiment, the target prompts are chosen to be the ones in I2P dataset [R2] as described in Section 4. Pertaining to the reviewer’s suggestion, we’ve added the corresponding definition in a notation table included in Appendix G.
>
> On the other hand, the intuition of linear combination has arisen from related literature [R1, R4]. Particularly, similar to our idea, [R1] attempts to create the steering activation vector by contrasting the activation of opposite prompts and appending them linearly to the user prompt for further manipulation. Meanwhile, [R4] attempts to model in-context learning via the so-called “in-context vectors” (ICV) that represent the specific task summary for a given language model. Furthermore, the ICVs could be utilized to fit the task that is aligned with the in-context demonstrations and the idea of learning multiple tasks or another way around could be done via simple arithmetics using these ICVs. **We believe that these ideas from the field of NLP have shared resonance with the linear combination approach of Ring-A-Bell to generate inappropriate images**.
>
> &nbsp;
>
> [R1] Turner, Alex, et al. "Activation addition: Steering language models without optimization." arXiv preprint arXiv:2308.10248 (2023).\
> [R2] Artificial Intelligence & Machine Learning Lab at TU Darmstadt. “AIML-Tuda/I2P · Datasets at Hugging Face.” AIML-TUDA/I2p · Datasets at Hugging Face, 20 Nov. 2022, huggingface.co/datasets/AIML-TUDA/i2p.\
> [R3] Wen, Yuxin, et al. "Hard prompts made easy: Gradient-based discrete optimization for prompt tuning and discovery." arXiv preprint arXiv:2302.03668 (2023).\
> [R4] Liu, Sheng, Lei Xing, and James Zou. "In-context Vectors: Making In Context Learning More Effective and Controllable Through Latent Space Steering." arXiv preprint arXiv:2311.06668 (2023).

---

> ### Author Response · Authors · 2023-11-20
> **Response for Reviewer UevV -- Part II**
>
> &nbsp;
>
> ### Ablation Study with Discrete Optimization
> ---
> We would like to clarify that discrete optimization is a necessary step in our black-box setting due to the discrete nature of the input text prompt (e.g., the word is not continuous), and we have included the ablation studies on different discrete optimization algorithms in Section 4.3.
> Specifically, The overall Ring-A-Bell is implemented based on the black-box assessment of current safety T2I diffusion models. That is to say, without the white-box access of T2I models, we are unavailable to input prompt signals in the form of soft prompts or embedding. As a result, it is only possible if we could construct a hard prompt (discrete) that is in latent space approximately to that of our original soft prompt $\mathbf{\tilde{P}}_{cont}$. As a result, the discrete optimization is **not an additional but instead an integral part of the Ring-A-Bell framework**.
>
> Lastly, we’ve experimented with different discrete optimization methods such as genetic algorithms and PeZ [R3] in the ablation study. We are happy to evaluate additional discrete optimization methods if the reviewer holds an interest in any other ones.
>
> &nbsp;
>
> ### Transferability Setting
> ---
> We thank the reviewer for raising the question. Indeed we’ve attempted to expand Ring-A-Bell to other possible text encoders such as BERT and T5. However, there are certain critical differences that hinder such encoders from achieving comparable performance against the CLIP-Based Ring-A-Bell. The main reason lies in that these text encoders specialize **in the text domain and construct respective embedding space, while the embedding space for CLIP is much more aligned from the vision-language aspect.** Secondly, the hard prompt provided by Ring-A-Bell starts from an original soft prompt specific to CLIP with further discrete optimization. This model-specific conversion could hinder the generated prompt to transfer under various text encoder settings.
>
> Nevertheless, we note that while the transferability is limited, in practice, **there is no reason to restrain the attacker to only use one text encoder such as CLIP.** We do not consider the differences in transferability as a weakness. **As long as the attacker can find one prompt to sabotage the target model, the attack is considered successful.** Therefore, the more text encoders the attacker can use, the more diverse prompts the attacker can create to test the target model.
>
> Moreover,  we note that most online T2I services would reveal not all but certain information about the model architecture or the underlying concept. This allows us to obtain the corresponding text encoder information for Ring-A-Bell to perform concept extraction. On the other hand, Ring-A-Bell could as well actively extract copies of certain concepts with possible text encoder candidates and apply the discrete optimization to acquire the respective hard prompt for evaluation.
>
> **Under this aspect, as long as a single problematic prompt from any text encoder has succeeded, this signifies the security breach of the corresponding of T2I models and its subsequent risk. Therefore we argue that the transferability of Ring-A-Bell does not hinder its potential in seeking possible risk of T2I models.**
>
> &nbsp;
>
> [R1] Turner, Alex, et al. "Activation addition: Steering language models without optimization." arXiv preprint arXiv:2308.10248 (2023).\
> [R2] Artificial Intelligence & Machine Learning Lab at TU Darmstadt. “AIML-Tuda/I2P · Datasets at Hugging Face.” AIML-TUDA/I2p · Datasets at Hugging Face, 20 Nov. 2022, huggingface.co/datasets/AIML-TUDA/i2p.\
> [R3] Wen, Yuxin, et al. "Hard prompts made easy: Gradient-based discrete optimization for prompt tuning and discovery." arXiv preprint arXiv:2302.03668 (2023).\
> [R4] Liu, Sheng, Lei Xing, and James Zou. "In-context Vectors: Making In Context Learning More Effective and Controllable Through Latent Space Steering." arXiv preprint arXiv:2311.06668 (2023).

---

> ### Author Response · Authors · 2023-11-22
> **Looking forward to reviewer UevV’s feedback**
>
> &nbsp;
>
> We appreciate the positive feedback and constructive comments from reviewer UevV in the initial reviews. As the discussion deadline nears, we haven't received additional feedback on our responses. We intend to use OpenReview's interactive feature to engage in discussions with the reviewer, confident that our response adequately addresses your concerns. Specifically, we've organized our response according to the reviewer’s suggestion as the following:
>
> * Introduction to Concept Removal Methods
> * Explanation of Prompt-Pair Selection and Generation
> * Explanation of Target Prompt and Intuition of the Linear Combination
> * Ablation Study with Discrete Optimization
> * Transferability Setting
>
> We hope our responses convince the reviewer about the merits of this work. If the reviewer has any other suggestions or comments, please don't hesitate to let us know!
>
> Best Regards, \
> Authors of Ring-A-Bell

---

### Author Response · Authors · 2023-11-20
**General Response**

We sincerely appreciate the reviewers for their precious feedback and insights on Ring-A-Bell. As there are many valuable comments from all reviewers that we wish to address, we’ve made some modifications pertaining to the structure of the overall paper listed in the following:
* The “What comes after explore?” paragraph has been moved from Conclusion to Appendix L.
* The notation table has been created in Appendix G for clarity. **(Reviewer UevV, Reviewer nnoZ)**
* Appendix H is included for the explanation of prompt-pair generations. **(Reviewer UevV)**
* Appendix I is included for the ablation study on different attack strategies concerning Ring-A-Bell. **(Reviewer WiiG)**
* Appendix J is included for additional experiments of smaller values of $K$. **(Reviewer WiiG)**
* Appendix K is included for the comparison against P4D. **(Reviewer nmRw)**
* Appendix M is included for the detailed discussion on similar works in assessing risk of T2I diffusion models. **(Reviewer nmRw)**

---

### Author Response · Authors · 2023-11-21
**Looking forward to the reviewer's feedback**

Dear reviewers,

We would like to start by thanking all reviewers for the positive feedback and constructive comments given in the initial reviews. As the discussion deadline approaches, we have yet to receive further feedback based on our responses. We would like to use the interactive feature of OpenReview to engage the reviewers in the discussion. In particular, in our responses, I believe we have provided additional discussions/clarifications and new experimental results to fully address the reviewer's concerns. We hope our responses convince the reviewers about the merits of this work. If the reviewer has any other suggestions or comments, please don't hesitate to let us know!

Best Regards,\
Authors of Ring-A-Bell

---

### Meta-Review · Area_Chair_69Fd · 2023-12-05

**Metareview:**

This paper studies the problem of red-teaming diffusion models for text-to-image synthesis (namely, transforming their prompts so that they evade existing safety mechanisms). It first formulates the problem as a model-specific optimization, where it assumes (unrealistically) that the attacker has (white-box) access to both the models with and without the safety mechanism. Then, the paper considers the (more realistic) model agnostic setting where the attacker only has black-box access to the model with the safety mechanism. It introduces an interesting optimization problem to find a modified prompt that tends to evade the safety mechanism. It runs an empirical evaluation which shows that the proposed method improves on an existing baseline.

One discussion that I feel is missing from the paper is whether one can design safety mechanisms that are resilient to the proposed attack.

Nevertheless, I feel that the topic is timely, and the paper interesting enough to justify acceptance.

**Justification For Why Not Higher Score:**

One discussion that I feel is missing from the paper is whether one can design safety mechanisms that are resilient to the proposed attack. Given the absence of such a discussion, it is unclear to me how much this paper will stand the test of time, and stay relevant even one year from now.

**Justification For Why Not Lower Score:**

I found the topic timely, and the paper interesting enough to justify acceptance.

---

### Decision · Program_Chairs · 2024-01-16

Accept (poster)